# Microstructure and Antibacterial Properties of Chitosan-Fe_3_O_4_-AgNP Nanocomposite

**DOI:** 10.3390/nano12203652

**Published:** 2022-10-18

**Authors:** Hartati Hartati, Subaer Subaer, Hasri Hasri, Teguh Wibawa, Hasriana Hasriana

**Affiliations:** 1Biology Department, Faculty of Mathematics and Natural Science, Universitas Negeri Makassar, Makassar 90222, Indonesia; 2Green of Excellence of Green Materials & Technology (CeoGM-Tech) FMIPA, Universitas Negeri Makassar, Makassar 90222, Indonesia; 3Physics Department, Faculty of Mathematics and Natural Science, Universitas Negeri Makassar, Makassar 90222, Indonesia; 4Chemistry Department, Faculty of Mathematics and Natural Science, Universitas Negeri Makassar, Makassar 90222, Indonesia

**Keywords:** nanoparticle, nanocomposites, antibacterial, conjugate, microstructure

## Abstract

The goal of this research is to synthesize and characterize Fe_3_O_4_@Chitosan-AgNP nanocomposites in order to determine their antibacterial activity. The research methods include the synthesis of Fe_3_O_4_@Chitosan-AgNP nanocomposites, as well as the characterization of nanoparticles using scanning electron microscopy (SEM), transmission electron microscopy (TEM), X-ray diffraction (XRD), Fourier transform infrared (FTIR) analysis, and subsequent antibacterial activity tests. The study’s findings demonstrated the successful synthesis of Fe_3_O_4_@Chitosan-AgNP nanocomposites, followed by nanoparticle characterization using SEM, TEM, XRD, and FTIR. Based on the XRD results, the conjugation of Fe_3_O_4_@Chitosan-AgNP nanocomposites has been successfully formed, as evidenced by the appearance of characteristic peaks of Fe_3_O_4_, chitosan, and AgNPs. According to the FTIR results, the interaction between chitosan-AgNPs and conjugated Fe_3_O_4_ occurred via the N atom in the NH_2_ group and the O atom in the OH group, and C=O. The SEM and TEM images also show that the Fe_3_O_4_@Chitosan-AgNP conjugation is a nanoparticle-based composite material. The combination of nanocomposites Fe_3_O_4_@Chitosan-AgNPs has antibacterial activity, inhibiting the growth of bacteria such as *Bacillus cereus* and *Escherichia coli*.

## 1. Introduction

Nanoparticles are intriguing materials to develop because their properties differ from those of other materials. As a result, numerous studies have been conducted to determine the potential of nanoparticles for use in a variety of applications. Precious metal nanoparticles have demonstrated significant potential for a wide range of applications in chemistry, biology, electronics, and catalysis [1]. Silver nanoparticles (AgNPs), as a fairly inexpensive precious metal, have been used in several catalytic reactions, such as selective ethanol oxidation, butadiene epoxidation, and NO_2_ reduction. This material is also widely used in other medical and scientific fields [2]. As a catalyst material, the active size of AgNPs and their dispersion in solution play an important role. The recovery of the extraction of AgNP catalysts from reaction mixtures becomes an important issue [3]. Thus, AgNPs doped with other solid materials (such as carbon, polymers, and metal oxides) receive special attention as composite catalysts with a stable structure [4]. 

Magnetite (Fe_3_O_4_) is an important type of magnetic material that has an inverted spinel cubic structure that has attracted attention due to its widespread use. Recently, Fe_3_O_4_ nanoparticles have emerged as promising catalyst supporters with high loading capacity, excellent dispersion, outstanding stability and convenient recycling. They help in improving the recovery and separation of nano-sized catalysts [5].

Chitosan is gaining popularity as a cationic polymer with biocompatibility, non-toxicity, antimicrobial, and even antitumor activity. Chitosan is a chitin deacetylation product [6]. Chitin is an industrial shrimp waste product that is widely produced throughout the world. Chitosan has a higher affinity for metal ions through chelation or ion exchange, which aids in nucleation during nanoparticle synthesis. There is currently no method for producing magnetically charged silver nanoparticles doped into chitosan (Fe_3_O_4_@Chitosan-AgNPs), which would result in magnetic nanocomposites derived from waste.

Nanoparticles of the precious metals silver (Ag) and gold (Au) have a role in various fields due to their unique physical and chemical properties, which are very different from their bulk particles [7]. The applications of these nanoparticles (NP) include such things as burn treatment, dental implantation materials, stainless coating materials, cosmetics, and biomedical applications, including as anti-cancer, antibacterial, and antioxidant agents. The role of a variety of diverse applications is related to the specific size and size distribution of nanoparticles. Several types of nanoparticles have been researched for biomedical applications, including copper carbonate nanoparticles for cancer cells [8], alginate acid nanoparticles for drug delivery [9], and zinc oxide nanoparticles as an antibacterial [10]. However, the antibacterial properties of Fe_3_O_4_@Chitosan-AgNPs have not been widely studied.

## 2. Materials and Methods

### 2.1. Synthesis of Fe_3_O_4_ Nanoparticles

We dissolved 4 g of FeCl_3_·6H_2_O in 125 mL of water with continuous stirring at 550 rpm and 90 °C. After 10 min, a clean solution with a yellow tint was produced. Subsequently, 2.7 g of FeSO_4_·7H_2_O was added to this yellow solution, and stirring continued for another 30 min. After that, 10 g of sodium hydroxide was dissolved in 40 mL of water, and it was added dropwise to the solution. The heating condition was maintained for another 1 h. After cooling, the magnetic field was used to separate the prepared black magnetic particles, followed by some washing of the deionized water. Finally, a free-flowing Fe_3_O_4_ nanoparticle was obtained after 24 h of air drying [1].

### 2.2. Synthesis of Fe_3_O_4_@Chitosan

A total of 0.75 g of chitosan was dissolved into 100 mL H_2_O. We added 2 wt% acetic acid to the mixed solution. We then added 1.5 g of Fe_3_O_4_ NPs to the solubility of the mixture, then in ultrasonics for 10 min. We continued the stirring for 20 min at room temperature at a speed of 500 rpm, then added 2 mL of 25 wt% glutardialdehyde at 40 °C for 3 h [11].

### 2.3. Synthesis of AgNPs

The bioreduction method was used to synthesize AgNPs from AgNO_3_ material, with *Terminalia catappa* leaf extract solution acting as the reduction agent. The synthesis of AgNPs with bioreductors from AgNO_3_ compounds, such as *Terminalia catappa* leaf extract solution, transforms Ag^+^ ions into AgO, which is stable [12]. With 100 mL of aquabidest, five grams of *Terminalia catappa* leaves powder were heated (boiling). After it had cooled, we filtered out the Whatman paper No.42. Following that, 20 mM AgNO_3_ was dissolved in 10 mL of *Terminalia catappa* leaf extract. We stirred the mixture at room temperature (60 min) (solution of bright purple color). 

### 2.4. Synthesis of Fe_3_O_4_@Chitosan-AgNPs Nanocomposites

We constantly stirred 50 mL of water milli-Q in a beaker and stored it in a water bath (60 °C). We then added 100 mg of chitosan-coated magnetic nanoparticles (Fe_3_O_4_@Chitosan). Ag solution (1 mL, 20 mM) was displaced, followed by the addition of 20 microliter sodium borohydride 0.1 M. Subsequently, 1 mL of ultra-filtered milk waste was added, and the solution changing from yellow to brown showed the formation of the Fe_3_O_4_@Chitosan-AgNP nanocomposites.

### 2.5. Characterization of Fe_3_O_4_@Chitosan-AgNPs Nanocomposite

A transmission electron microscope was used to further characterize the size and morphology of the resulting sample (TEM). X-ray diffraction was used to investigate the crystalline properties of the nanocomposite (XRD). Using a scanning electron microscope, we examined images of the nanocomposite samples (SEM). Fourier transform infrared (FT-IR) spectroscopy revealed surface groups and covalent bonds among chitosan, Fe_3_O_4_, and AgNPs.

### 2.6. Antibacterial Activity

The antibacterial activity of conjugated Fe_3_O_4_@Chitosan-AgNPs was determined by the diffusion method using nutrient agar (NA) medium. This test uses two types of bacteria, namely *Escherichia coli* (Gram-negative bacteria) and *Bacillus cereus* (Gram-positive bacteria). Fe_3_O_4_@Chitosan-AgNP samples with a concentration of 100 μg/mL were used to test the activity of bacterial inhibition zones. The positive control used chloramphenicol (30 μg/mL). The determination of sample concentration refers to the research of Thukkaram et al. [13]. After 24 h, the diameter of the inhibitory zone on the petri dish was measured. The diameter of the inhibition zone indicates the antibacterial activity.

## 3. Results

### 3.1. Characterization of Fe_3_O_4_@Chitosan-AgNPs

#### 3.1.1. XRD Characterization (X-ray Diffraction)

The results of XRD characterization of Fe_3_O_4_@Chitosan-AgNPs of nanocomposites can be seen in Figure 1. For different compositions, the peak of each constituent is clearly discernible. The diffractogram confirms the formation of nanocomposites. The highest peak of Fe_3_O_4_NPs is observed with a crystallographic plane (311) at 2θ = 37.99° (PDF#01-072-6170). The peak of AgNPs with a crystallographic plane of (004) is observed at 2θ = 35.28° (PDF#01-087-0598), and the highest peak for chitosan is observed at 2θ = 31.61° (PDF#00-055-16110). The amorphous nature of chitosan decreases as the amount of Fe_3_O_4_ and AgNPs increases.

#### 3.1.2. SEM Characterization (Scanning Electron Microscope)

Figure 2 depicts the SEM-EDX characterization results. EDX Mapping-SEM Confection Fe_3_O_4_@Chitosan-AgNPs results. Table 1 shows the percentage of Fe_3_O_4_@Chitosan-AgNPs connotation elements. 

#### 3.1.3. Characterization of FT-IR (Fourier Transform Infrared)

FT-IR characterization is used to identify organic, polymeric and some inorganic materials (Figure 3). The method of analyzing the composition of the product with FT-IR uses infrared rays to scan the test sample and observe its chemical properties. 

#### 3.1.4. Characterization of UV-Vis

Visible light spectrophotometry (UV-Vis) is the measurement of light energy by a chemical system at a certain wavelength. Ultraviolet (UV) light has a wavelength between 200 and 400 nm, and visible light has a wavelength of 400–750 nm. Spectrophotometric measurements use a spectrophotometer that involves considerable electronic energy in the analyzed molecules, so UV-vis spectrophotometers are more widely used for quantitative analysis than qualitative.

Figure 4 shows that the absorption spectrum of UV-Vis of Fe_3_O_4_@Chitosan-AgNP nanocomposites in the range of 200–400 nm was measured by a UV-1280 UV VIS (Shimadzu) spectrophotometer operated in dual beam mode using a quartz cuvette with a wavelength of 1.0 cm optical line length. The UV-vis spectrum reveals that the surface plasmon resonance band (PRB) of the Fe_3_O_4_@Chitosan-AgNP nanocomposite is composed of several peaks originating from the PSB of each component. The surface plasmon resonance (SPR) of Fe_3_O_4_@Chitosan-functionalized AgNPs is responsible for the highest peak at around 236 nm. The results of UV-vis characterization served as a benchmark for the stability of nanocomposites for a long time [14,15]. Other peaks were formed as a result of the absorbance of Fe_3_O_4_NPs between 200 and 300 nm [16].

#### 3.1.5. TEM (Transmission Electron Microscopy) Characterization

To obtain nanostructure information from the Fe_3_O_4_@Chitosan-AgNP conjugation sample, testing was performed using a transmission electron microscope (TEM), Tecnai G2, at 50,000–200,000 times magnification. Powders from the conjugation of Fe_3_O_4_@chitosan-AgNP (0.2) nanocomposites and Fe_3_O_4_@chitosan-Ag (0.4) nanocomposites were tested. These are hybrid materials composed of chitosan organic polymer matrix and metal-based nanoparticles Fe_3_O_4_NPs and AgNPs. Figure 5 and Figure 6 show the TEM images of the Fe_3_O_4_@Chitosan-AgNP conjugation.

Figure 5 depicts a TEM image of the Fe_3_O_4_@Chitosan-AgNP (0.2) sample at two different magnifications, with scale bars of 10 nm and 5 mm, respectively. The two TEM images revealed that Fe_3_O_4_ nanoparticles with particle sizes between 10 and 20 nm had a relatively spherical but not perfect shape, followed by AgNPs with particle sizes smaller than 10 nm and darker images (grayscale) because the atomic number of Ag was greater than the atomic number of Fe. Chitosan, as an organic polymer, is a matrix that contains the two nanoparticles, Fe_3_O_4_NPs and AgNPs, which appear as a thin layer on the surface of the nanoparticles. 

The surface of Fe_3_O_4_NPs shows a regular Fe crystal lattice pattern at a magnification of 200,000 times (scale bar 5 nm). This is a result of energetic electrons from the TEM filament interacting with atoms in the Fe crystal lattice. The TEM image of this sample indicates that the Fe_3_O_4_@Chitosan-AgNP conjugation is a nanoparticle-based composite material.

Figure 6 shows a TEM image of the Fe_3_O_4_@Chitosan-AgNP (0.4) sample with a relatively uniform and spherical Fe_3_O_4_NP grain size between 10 and 20 nm and AgNP grains smaller than 10 nm. In both images, chitosan material forms a coating on the surface of larger nanoparticles, causing the grayscale of the particles to be uneven. Fe_3_O_4_ particles are very dark and distinct from other particles at a scale bar of 5 nm. The chitosan layer on the surface of the nanoparticles could be very thick and cover the entire surface of the particles. At very high magnifications, a Fe crystal lattice can be seen, indicating the high quality of the highly synthesized material. 

In order to demonstrate the distribution of Fe_3_O_4_NPs and AgNPs conjugated with chitosan, we provide SEM images for Fe_3_O_4_@Chitosan-AgNPs (0.2) and Fe_3_O_4_@Chitosan-AgNPs (0.4) samples, as well as X-ray mapping, as shown in Figure 7. The X-ray mapping indicated that both nanoparticles, Fe_3_O_4_NPs and AgNPs, were distributed evenly throughout the surface of chitosan, indicating a strong chemical bond in the formation of the nanocomposites. 

### 3.2. Antibacterial Properties of Fe_3_O_4_@Chitosan-AgNPs

Antibacterial activity of Fe_3_O_4_@Chitosan-AgNPs samples can be seen in Figure 8 and Figure 9. Statistical analysis revealed that samples of Fe_3_O_4_@Chitosan-AgNPs can significantly inhibit the growth of *B. cereus* and *E. coli* bacteria, with effects comparable to positive controls (Chloramphenicol). The high inhibition zone formed in the activity test on *B. cereus* (Gram-positive bacteria) and *E. coli* indicated this result (Gram-negative bacteria). This demonstrates that the sample is an excellent antibacterial material. Chloramphenicol is a commercial antibiotic that is used to treat bacterial infections. The bacteria *E. coli* and *B. cereus* are two of several pathogenic bacteria that are often the causative agents of foodborne disease.

## 4. Discussion

The results of XRD characterization of Fe_3_O_4_@Chitosan-AgNPs of nanocomposites can be seen in Figure 1. This result shows the diffraction pattern of Fe_3_O_4_ at position 2θ = 30°–31°, nano chitosan at position 2θ = 20°–30°, and AgNPs at position 2θ = 35°. Based on the results obtained, we can state that the conjugation of Fe_3_O_4_@Chitosan-AgNP nanocomposites has been successfully formed and is characterized by the appearance of characteristic peaks of Fe_3_O_4_, chitosan and AgNPs.

EDX profiles of Fe_3_O_4_@Chitosan–AgNPs nanocomposites were used to calculate the mass fraction of element mapping. Figure 2 of the obtained EDX profile clearly depicts the silver peak being at 3 keV and iron at between 6 and 8 keV for the Ag and Fe_3_O_4_NPs. Quantitative element analysis data obtained that Fe_3_O_4_@Chitosan–Ag nanocomposite contains 29.50% (*w*/*w*) Ag. Structural information on Fe_3_O_4_@Chitosan–Ag nanocomposites with FE-SEM analysis is shown in Figure 2. The white dots on the nanocomposites indicate the presence of Ag nanoparticles on the surface. The results of the element mapping showed that the elements C, Fe, and Ag were evenly distributed across the prepared nanocomposites, indicated by different colors that confirmed the uniform distribution of Ag nanoparticles on the surface of Fe_3_O_4_@Chitosan–Ag nanocomposites. Table 1 shows that the presence of components C, N, O, Fe and Ag with their mass percentages, respectively, proves the success of the consignment of Fe_3_O_4_@Chitosan–AgNP nanocomposites.

FT-IR spectra are used to determine the existence of functional groups of each material before and after the addition of other materials according to their designation. The analysis of functional groups in Figure 3 can be described as follows. For chitosan (blue color), the spectra at wave number 3453 cm^−1^ indicates the presence of the -OH (hydroxyl) and -NH_2_ (amin) groups overlapping each other so that the spectra appear to widen. The wave number 2959 cm^−1^ indicates that the C-H group is supported by the 1401 cm^−1^ C=C aromatic C=C, and the aromatic C-H of chitosan is supported in the fingerprint area 873; 675; and 591 cm^−1^,1654 cm^−1^ indicates the carbonyl groups C-O and or C-O-C are supported at wave numbers 1320 and 1242 cm^−1^.

After the addition of a silver nanoparticle material (AgNPs), Fe_3_O_4_ conjugation showed that the wave number of 3400 cm^−1^ did not appear due to the interaction of transition metal cations with chitosan, forming metallopolymers [Ag-Chi]^+^. Nevertheless, a typical functional group appeared in the fingerprint area of 873 cm^−1^. Similarly, there was a shift in the wave numbers in the C-H group from 2959 cm^−1^ to 2924 cm^−1^, the emergence of new spectra at wave numbers 1701 cm^−1^, 1763 cm^−1^ and 1636 cm^−1^ indicating C=O (NHCOCH_3_-), and wave numbers at 1544 cm^−1^ and 1384 cm^−1^ indicating NH (R-NH_2_) could it was concluded that the interaction between chitosan-AgNP conjugation Fe_3_O_4_ had occurred through the N atom in the NH_2_ group and the O atom on the OH group, as well as C=O.

The TEM image, in Figure 5, for the Fe_3_O_4_@Chitosan-AgNP (0.2) sample is shown with two different magnifications with scale bars of 10 nm and 5 mm, respectively. The two TEM images showed the dominance of Fe_3_O_4_ nanoparticles with particle sizes between 10 and 20 nm with a relatively spherical, although not perfect, shape; it also shows AgNPs with particle sizes smaller than 10 nm and with darker images (grayscale) because the atomic number of Ag was greater than the atomic number Fe. As an organic polymer, chitosan is a matrix that includes the two nanoparticles Fe_3_O_4_NPs and AgNPs, which appear to be a thin layer on the surface of the nanoparticles. At a magnification of 200,000 times (scale bar 5 nm), the surface of Fe_3_O_4_NPs shows a regular Fe crystal lattice pattern. This is an indication of the interaction of energetic electrons from the TEM filament with atoms in the Fe crystal lattice. The TEM image of this sample is an indication that the conjugation of Fe_3_O_4_@Chitosan-AgNPs is a nanoparticle-based composite material.

The TEM image in Figure 6 for the Fe_3_O_4_@Chitosan-AgNP (0.4) sample shows a relatively uniform Fe_3_O_4_NP spherical grain size between 10 and 20 nm, as well as AgNP grains smaller than 10 nm. In both images, there is no chitosan material that becomes a coating on the surface of larger nanoparticles, which makes the grayscale of the particles not uniform. At a scale bar of 5 nm, there are Fe_3_O_4_ particles that are very dark and different from other particles. It is possible that the chitosan layer on the surface of the nanoparticles is very thick and covers the entire surface of the particles. At very large magnifications, a Fe crystal lattice also appears, which is an indication of the quality of the highly synthesized material.

The effect of Fe_3_O_4_@Chitosan-AgNP samples in inhibiting the growth of *B. cereus* bacteria can be seen in Figure 8. Based on statistical analysis, it was shown that this sample had the same effect as the positive control (chloramphenicol) in inhibiting the growth of *B. cereus* bacteria. Figure 9 shows that samples of Fe_3_O_4_@Chitosan-AgNPs statistically also had the same effect as positive controls in inhibiting the growth of *E. coli* bacteria. This informs that samples of Fe_3_O_4_@Chitosan-AgNPs can be used as a consignment nanoparticle material that is antibacterial. The results of previous studies show that chitosan functionalized Ag nanoparticles have a great bactericidal efficiency against bacteria and fungi [13]. AgNP and AuNPs are small-sized materials, but their availability in large quantities is important because they are widely needed for commercial and industrial applications. The synthesis of AgNPs with bioreductors from AgNO_3_ compounds, such as *Terminalia catappa* leaf extract solution, produces Ag^+^ ions to AgO which is stable [12]. AgNPs have also been widely applied in sterilization due to their excellent antibacterial properties. However, AgNPs require good storage conditions as well because their antibacterial performance is strongly influenced by environmental conditions. Zang et al. explained that triboelectric nanogenerator (TENG)-based electrodeposition as a self-powered source can synthesize finer-sized AgNPs (<10 nm) with outstanding antibacterial effects against *E. coli* and *S. aureus* with 100% efficiency at 2 h [17].

The results of this study showed that Fe_3_O_4_@Chitosan-AgNP samples had a significant effect in inhibiting the growth of *B. cereus* bacteria as Gram-positive bacteria and *E. coli* as Gram-negative bacteria. The difference between these two types of bacteria is that there are different cell wall constituents. These results show that Fe_3_O_4_@Chitosan-AgNPs can be developed as nanomaterials that are antibacterial. Previous studies have reported that Fe_3_O_4_NPs at a concentration of 120 µg/mL had no cytotoxicity effect on normal line Hs68 cells with 89% viability [18]. Zhang et al. have also reported that although magnetic Fe_3_O_4_ nanoparticles at sizes 120 and 250 nm at various concentrations do not have significant toxicity in chicken macrophage cells (HD11). However, the cytotoxicity of these nanoparticles increases with decreasing size [19]. Thus, previous studies have revealed that the cytotoxicity effects of these nanoparticles are related to their size, concentration, time, shape and cell type [19,20,21]. 

Other studies have shown that the antibacterial activity of Fe_3_O_4_NPs at concentrations of 0.01, 0.05, 0.1, and 0.15 mg/mL has a significant effect in inhibiting the growth of the bacteria *Pseudomonas aeruginosa*, *Escherichia coli*, and *Staphylococcus aureus* [13]. Gabrielyan et al. have also reported antibacterial activity in vitro of silver (Ag)- and citric acid-coated Fe_3_O_4_NPs against wild-type *E. coli* and kanamycin-resistant strains, as well as *Salmonella typhimurium* MDC1759. Their results showed a strong antibacterial effect on the dose-dependent response of the tested bacteria through changes in membrane permeability and membrane bonds on enzyme activity [22]. Armijo et al. also revealed the antimicrobial effect of iron oxide nanoparticles at concentrations of 1.78–17.35 mg/mL can inhibit *Pseudomonas aeruginosa* [23].

## 5. Conclusions

The conjugation of Fe_3_O_4_@Chitosan-AgNP nanocomposites has been successfully formed and characterized by the appearance of characteristic peaks of Fe_3_O_4_, chitosan, and AgNPs. The interaction of chitosan-AgNPs conjugation Fe_3_O_4_ occurred via the N atom in the NH_2_ group and the O atom in the OH group, as well as C=O. The result also indicates that the conjugation of Fe_3_O_4_@chitosan-AgNPs is a nanoparticle-based composite material. The conflation of Fe_3_O_4_@Chitosan-AgNP nanocomposites has antibacterial activity by inhibiting the growth of bacteria such as *Bacillus cereus* and *Escherichia coli*.

## Figures and Tables

**Figure 1 nanomaterials-12-03652-f001:**
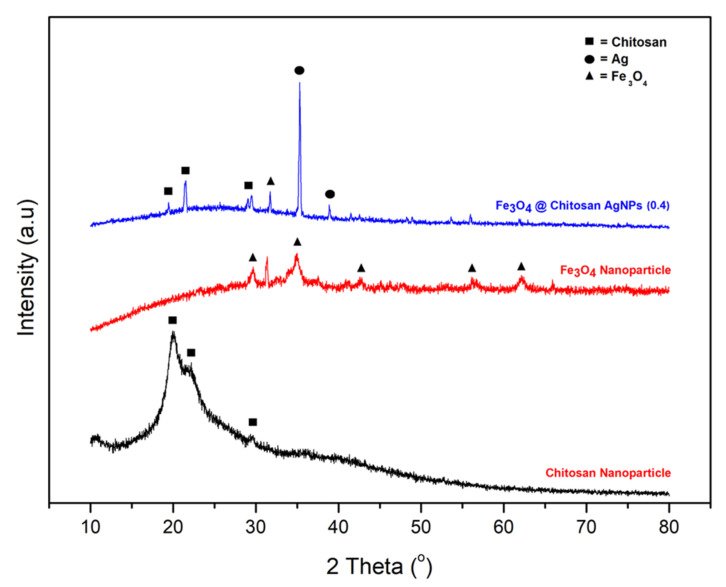
X-ray diffractogram pattern of chitosan, Fe_3_O_4_ nanoparticles and Fe_3_O_4_@Chitosan-AgNP nanocomposites.

**Figure 2 nanomaterials-12-03652-f002:**
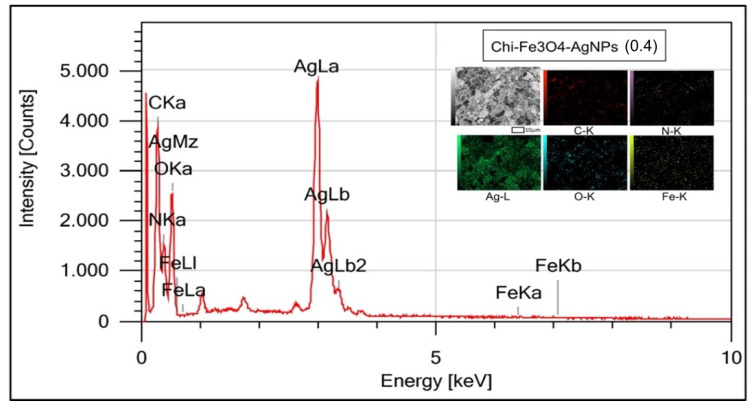
Results of EDX Mapping-SEM conjugation Fe_3_O_4_@Chitosan-AgNPs.

**Figure 3 nanomaterials-12-03652-f003:**
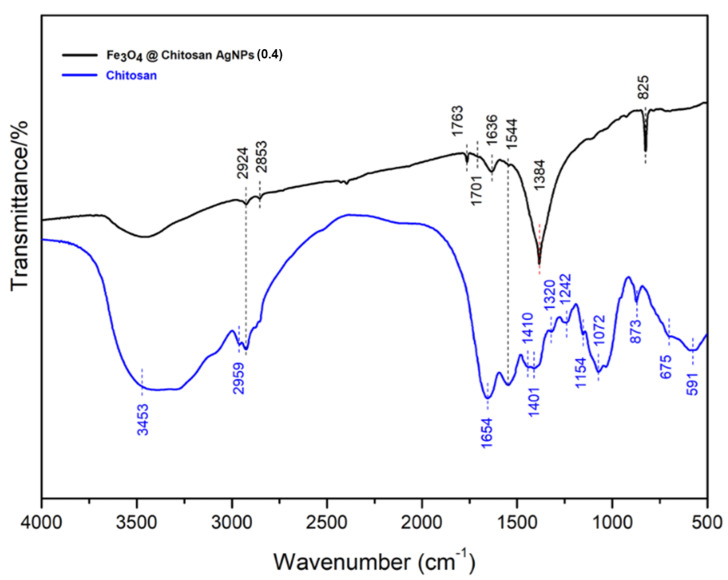
FTIR spectrum of Fe_3_O_4_@Chitosan-AgNP nanocomposites.

**Figure 4 nanomaterials-12-03652-f004:**
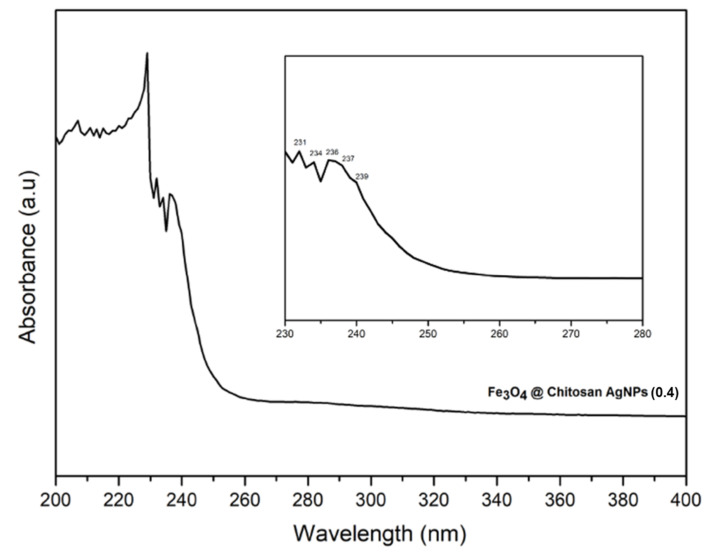
UV-visible spectra of Fe_3_O_4_@Chitosan-AgNP nanocomposites.

**Figure 5 nanomaterials-12-03652-f005:**
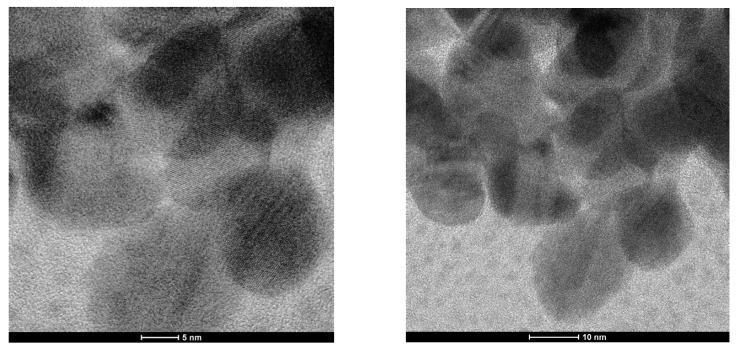
TEM image of Fe_3_O_4_@Chitosan-AgNPs (0.2).

**Figure 6 nanomaterials-12-03652-f006:**
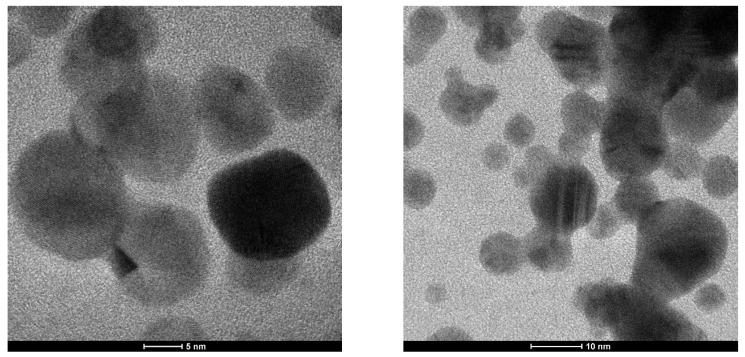
TEM image of Fe_3_O_4_@Chitosan-AgNPs (0.4).

**Figure 7 nanomaterials-12-03652-f007:**
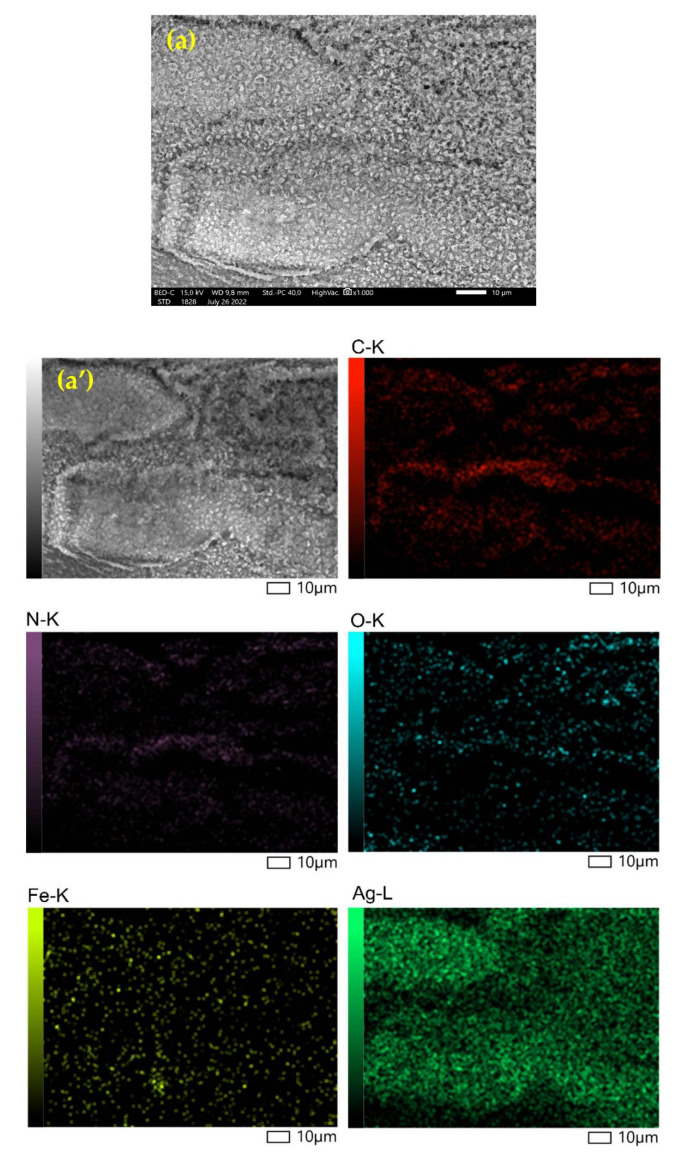
SEM image and X-ray mapping for (**a**,**a’**) Fe_3_O_4_@Chitosan-AgNPs (0.2) and (**b**,**b’**) Fe_3_O_4_@Chitosan-AgNPs (0.4) showing the distribution of Fe_3_O_4_NPs and AgNPs conjugated with chitosan.

**Figure 8 nanomaterials-12-03652-f008:**
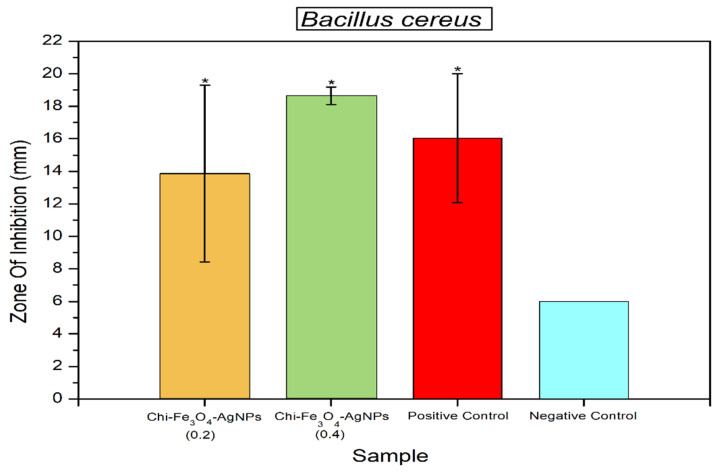
Antibacterial activity of Fe_3_O_4_@Chitosan-AgNPs against *B. cereus*. Mean ± SD (*n* = 3); * *p* < 0.05 significant change compared with the negative control group.

**Figure 9 nanomaterials-12-03652-f009:**
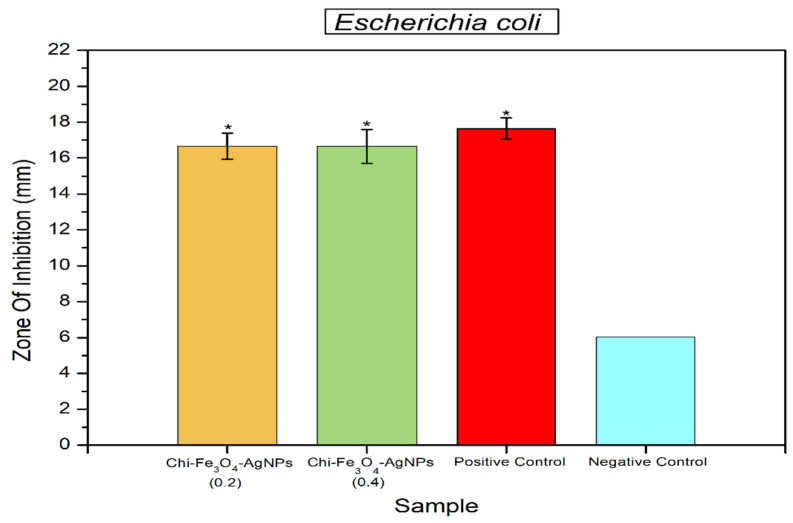
Antibacterial activity of Fe_3_O_4_@Chitosan-AgNPs against *E. coli*. Mean ± SD (*n* = 3); * *p* < 0.05 significant change compared with the negative control group.

**Table 1 nanomaterials-12-03652-t001:** Percentage of connotation elements Fe_3_O_4_@Chitosan-AgNPs (0.4).

Element	Mass (%)	Atom (%)
C	20.30 ± 0.07	32.19 ± 0.11
N	17.01 ± 0.18	23.14 ± 0.24
O	33.13 ± 0.26	39.44 ± 0.31
Fe	0.06 ± 0.02	0.02 ± 0.01
Ag	29.50 ± 0.16	5.21 ± 0.03
Total	100.00	100.00

## Data Availability

Not applicable.

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
