# Peer review of "Microstructure and Antibacterial Properties of Chitosan-Fe3O4-AgNP Nanocomposite"

_nanomaterials, 2022, doi:10.3390/nano12203652_

Round 1

Reviewer 1 Report

There are some problems need to be considered. The following points should be added/changed to further improve:

The writing of Fe3O4 is inconsistent throughout the text, please standardize the writing. The manuscript is written in a very poor format. 

  1. Introduction: Please consider the following articles in your review of the state of the art (DOI: 10.3390/toxics10030144). No mention of Fe3O4, why Fe3O4 is needed for this manuscript and what role it plays.
  2. Materials and Methods: In line 67, "2.7 g of FeSO4.7H2O was added and stirred", please standardize the reagents. What is the purpose and significance of using bioreduction to synthesize AgNPs? What are the advantages over other means of synthesizing AgNPs?
  3. Materials and Methods: In line 100, "Each sample used a concentration of 10% was dripped on 100 paperdish. Positive control using Chloramphenicol" What is the concentration of chloramphenicol? How does it compare with NPs? Why sample used a concentration of 10%?
  4. Results and Discussion: The writing format is not uniform, please standardize. In Figure 1, please add the standard PDF for cross-referencing. What is the purpose and meaning of the UV-vis in Figure4 in this work? There is no evidence of the role and meaning of UV-vis in this manuscript. In Figure5 and Figure6, please provide HRTEM to illustrate the structure and location of Fe3O4 NPs, AgNPs and Fe3O4@chitosan-AgNPs. There was no evidence that the antibacterial originated from Ag/Fe/Chitosan, and controlled trials were lacking.
  5. The inhibition experiments were done too simply and the inhibition circles made from just one experimental concentration are not sufficient to tell the story. More experiments need to be added.
  6. Discussion: Does Fe3O4 NPs particle size affect toxicity? Please include this aspect in your discussion.

Author Response

Thank you to reviewer 1 for all your valuable corrections and suggestions. We have included them in our new manuscript. Please find our response in the attachment file.

Reviewer 2 Report

There are so much question in this manucript, the authors should be carefully revised the manusctipt before further consideration. The specific questions are as follows

1.  Many superscript/subscript used unreasonable.

2. The name of material isn't uniform, such as Fe2O3/Fe3O4;  Fe3O4@chitosan-AgNPs/ Fe3O4-Chitosan-AgNPs nanocomposites, ect.

3. In expertial section, many typos can be seen, such as FeSO 4.7H 2 O, ect.

4. In figure 1 , the standard card shoud be added.

5.   Fe3O4-Chitosan-AgNPsNanocomposite should be characterizated by single nanoparticle mapping.

6. both B. cereus and E. coli are gram negative bacteria, the antibacterial activity toward gram positive should be added.

7. Some references should be cited in the manuscript ,such as DOI: 10.1021/acs.jpclett.2c01737 ; 10.1002/smll.202000436.

Author Response

Thank you very much to reviewer 2 for you important corrections and suggestions. We have revised the manuscript based on all your corrections and suggestions. 

Reviewer 3 Report

The authors have done a very interesting study on Chitosan-Fe3O4- 2 AgNPs Nanocomposite for its antibacterial activity. The synthesized nanoparticle is well characterized and studied using various instrumentations such as XRD, EDAX, FTIR, TEM etc. The work is indeed very interesting and has a promising outcome. However there are some minor comments which I would like the authors to address.

1. The opening sentence of both Abstract and the Introduction are very similar and looks repetitive. It would be good to improve the opening sentences.

2. In the introduction the authors have discussed about chitin etc, rather from this article point of view the authors could discuss a similar article where chitosan is used with metallic nanoparticles for biomedical applications (https://doi.org/10.1039/c5tx00212e). Authors could discuss these articles to highlight the importance of chitosan here.

3. Similar to chitosan various other natural biopolymers have been studied for antibacterial activity. As the authors discuss the use of silver nanoparticles in their later half of introduction, alginic acid is another such biopolymer as chitosan which is used along with silver to synthesise nanoparticles for antibacterial activity (doi.org/10.1007/s12257-010-0099-7). The author could discuss this in their introduction or discussion section which is relevant to their work. 

4. Is there any specific reason why the authors have decided to abbreviate chitosan-coated magnetic nanoparticles as “Kitosan” such as in line 84, 97 etc? If not, then it would be good to use constant spelling of chitosan throughout the manuscript.

5. The section 2.4 i.e. line 54 the spelling of “Nanocomposit” should be corrected.

6. What does the author mean by the word “Conceivation” in the figure cation of Fig5. It could simple be written as “TEM image of Fe3O4@Chitosan-AgNPs”.

7. In figure 7, the authors have used alphabets such as “ab, b, a” on the bar graphs, but failed to mention them in the figure caption. Same in figure 8.

Author Response

Thank you very much to reviewer 3 for your kind corrections and suggestions. We have accommodated all your corrections and suggestions to our revised manuscript.

Round 2

Reviewer 1 Report

No further suggestion needed. It can be accepted and published in present format. 

Author Response

Dear reviewer 1

Thank you for accepting our response and recommend it to publish our manuscript.

Reviewer 2 Report

Figure 7 and 8 sould be improved, and the Y axis should be presented.

Author Response

Dear reviewer 2.

Thank you very much for your suggestion. Please find our response.
